# Effect of Different Additives on the Quality of Rehydrated Corn Grain Silage: A Systematic Review

**Luciana Viana Diogénes** [1,*], **José Morais Pereira Filho** [1], **Ricardo Loiola Edvan** [2],
**Juliana Paula Felipe de Oliveira** [3], **Romilda Rodrigues do Nascimento** [1], **Edson Mauro Santos** [4],
**Elisvaldo José Silva Alencar** [5], **Pedro Henrique Soares Mazza** [6], **Ronaldo Lopes Oliveira** [6]
and **Leilson Rocha Bezerra** [1,*]

1  Department of Animal Science, Federal University of Campina Grande, Patos 58708110, Paraíba, Brazil;
   jmpfpiaui@gmail.com (J.M.P.F.); romildarn01@ufpi.edu.br (R.R.d.N.)
2  Animal Science Department, Federal University of Piaui, Teresina 64049550, Piaui, Brazil; edvan@ufpi.edu.br
3  Animal Science Department, Federal University of Sergipe, Nossa Senhora da Glória 49680000, Sergipe, Brazil;
   jupaula.oliv@yahoo.com.br
4  Center of Agrarian Sciences, Federal University of Paraiba, Areia 58397000, Paraíba, Brazil; edson@cca.ufpb.br
5  Forage Department, Federal University of the San Francisco Valley, Petrolina 40170110, Pernambuco, Brazil;
   johnny.alencar@hotmail.com
6  Department of Animal Science, Federal University of Bahia, Salvador 40170115, Bahia, Brazil;
   pedromazza@outlook.com (P.H.S.M.); ronaldooliveira@ufba.br (R.L.O.)
*  Correspondence: luhvianadiogenes@hotmail.com (L.V.D.); leilson@ufpi.edu.br (L.R.B.)

**Abstract:** This review aimed to analyze the effects of additives in producing silage from rehydrated corn grains for ruminants. The control treatment studies used in this analysis involved corn grain rehydrated with water only. To be included in the review, the studies needed to follow standardized criteria, including the absence of additives in the control treatment and the silage evaluation of the in animals such as cattle, goats, and sheep. A total of fifteen publications between 2014 and 2023 were included in the final dataset. The PROC ANOVA of SAS was used to compare the results, which included a random effect of comparison within the study, performing a paired comparison. It was observed that additives did not influence the chemical composition, pH, organic acid, ethanol content, microbial population, fermentative losses, aerobic stability, and dry matter in vitro digestibility of rehydrated corn grain silage ($p > 0.05$). Using additives in corn silage is a promising practice that can significantly benefit silage fermentation. Moisture silage additives mitigate high mycotoxin levels, enhance aerobic stability, improve cell wall digestibility, and increase the efficiency of utilization of silage nitrogen by ruminants. Using fermentation-stimulating additives (*Lactobacillus buchneri*) can improve the quality of rehydrated corn grain silage. There are still a few studies and more research to elucidate the best additives and the ideal amount to be added to ground corn grain silage.

**Keywords:** additives; corn grains; digestibility; silage





## 1. Introduction

Livestock production in tropical regions, including Brazil, faces challenges in grain availability due to climatic conditions that affect production. To circumvent this situation, it is possible to adopt strategies to increase animal production efficiency [1–3]. Among these strategies, rehydration and ensiling of corn grains have significant advantages, such as reducing storage costs and improving grain utilization efficiency [4–8].

The predominant cultivation of corn hybrids in Brazil, with a higher proportion of vitreous endosperm [9–11], is negatively related to starch digestibility [12–14], making ensiling an advantageous practice in terms of storage management and nutritive value [15–17]. In addition, ensiling reduces insect and rodent damage normally seen in dry grains and increases starch digestibility [18–20].

Corn grain silages and other grains have shown excellent fermentation processes due to the degradation of prolamin. Proteolysis is an undesirable process in most silages, as it increases ammonia nitrogen concentrations and compromises animal performance. However, corn grain silage is desirable for a certain degree of proteolysis, as it promotes the degradation of the protein matrix, increasing nutrient availability [21,22].

Ensiling time gradually influences the amount of these prolamins. These proteins are insoluble in water and rumen fluid. However, they can be solubilized in an acidic environment during anaerobic fermentation. Therefore, to maximize the degradation of the protein matrix and improve the use of nutrients, a fermentation period of more than 50 days is recommended because this can maximize the availability of carbohydrates [23,24].

In addition, it is important to consider that the specific composition of hydrophobic proteins (zeins) and the presence of other components of corn and the silage environment can have variable effects on the digestion process in ruminants. Further studies employing advanced analytical techniques, such as mass spectrometry and molecular biology techniques, can provide crucial information about the fate and impact of prolamins during the ensiling process [25,26].

In ruminant feeding, when ensiling corn grain silage, it is necessary to observe some crucial points to ensure the product's final quality. During crop development, the soluble carbohydrates present in the grains are polymerized into starch in the endosperm, which results in small amounts of readily fermentable carbohydrates [27], the main substrates for the growth of lactic acid bacteria. These bacteria are responsible for the rapid acidification of the ensiled mass [28]. In addition, corn grain silages are more likely to suffer aerobic deterioration due to their starch content [29]. To minimize these adverse effects and improve silage quality, additives are expected to increase the content of soluble carbohydrates, reduce fermentation losses, and increase aerobic stability during the ensiling process.

Additives are substances added at the time of ensiling that aim to stimulate lactic fermentation, inhibit fermentation by undesirable microorganisms, and, consequently, reduce fermentative losses, which can improve the nutritional value of the silage [30]. Furthermore, using additives in ensilage can promote improvements in aerobic stability, increased consumption, and animal performance. However, hardly any additives have all these characteristics [31].

Thus, to obtain silages with an adequate fermentative profile, the plant must present some characteristics inherent to its chemical composition, such as a dry matter content of approximately 30%, a soluble carbohydrate content of 10%, and a low buffering capacity (20 mg of NaOH/100 g DM). However, most plants do not present these prerequisites, making it essential to use additives that stimulate fermentation [32].

According to Tian et al. [33], additives can be classified into four categories: fermentation inhibitors (malic acid), fermentation accelerators (glucose), cellulase (enzymes), and microbial inoculant (*Lactobacillus buchneri*). Among the additives that stimulate lactic fermentation, it is possible to highlight the use of microbial inoculants. These are environmentally friendly, easy to apply, and, therefore, most used in the ensiling process [32,34]. On the other hand, enzymatic additives are protein compounds that promote chemical reactions. These additives can be combined with microbial inoculants to increase the number of substrates for lactic acid bacteria [35].

According to Nolan et al. [36], enzymatic additives act on fiber hydrolysis, promoting increased availability of soluble carbohydrates for lactic acid bacteria. However, the importance of rapid action of the additive is highlighted, as high temperatures can inactivate the action of enzymes. Therefore, to guarantee the effectiveness of enzymatic additives, it is necessary to observe the enzyme's mode of action [37]. According to these authors, enzymatic activity should promote hydrolysis inside the silo, provide substrates for lactic acid bacteria, and increase enzymatic activity in the rumen. Therefore, for fiber degradation to occur, the action of several enzymes from the additive, the plant, and rumen microorganisms is necessary. In this sense, the degradation process consists of the enzyme adhering to the substrate and, subsequently, the partial degradation of the fibrous constituents [38].

The relevance of this Investigation is highlighted due to the importance of these grains in animal production. This review hypothesizes that adding additives when ensiling rehydrated or reconstituted corn grains can promote improvements in silage quality, increase starch digestibility, and reduce fermentative and aerobic losses. This improvement in silage quality can result in increased animal performance, presenting a viable alternative for using these grains in feeding high-producing ruminants. Thus, the objective of this study is to perform a systematic review of the scientific literature to investigate the effects of different additives in the ensiling of rehydrated or reconstituted corn grains, evaluating the quality of the silage and its impact on animal performance, as well as comparing the different insights observed from the research carried out on the topic.

## 2. Materials and Methods

### 2.1. Dataset

This systematic review was carried out according to the Preferred Reporting Items for Systematic Reviews and Meta-Analyses (PRISMA) guidelines [39]. For example, items such as the title in the PRISMA checklist indicate that it is a systematic review. The objectives include a straightforward question, i.e., containing the people or problem that will be addressed in the review, the type of intervention that will be analyzed, whether there will be a comparison between different interventions, and what the results (outcomes) analyzed in the selected studies show; there is an indication that the systematic review follows a protocol and was registered on a review platform; indicates the criteria for inclusion of studies in the review; indicates which sources of information were used, including at least one bibliographic database, and explains how the search strategies used were carried out; and explains how the reading process and application of the selection criteria were carried out.

The PICO strategy followed the PRESS guidelines statement [40] and defined the population as beef cattle, sheep, and goats, the intervention as the evaluation of the effect of additives on silage quality and animal performance, the control as silages without additives, and the outcomes as the most suitable additives to improve silage quality.

Database searches were performed between January and November 2023, based on title and abstract, and with language refinement, including articles in English and Portuguese. The following databases were used for the literature search: Web of Science, Wiley Online Library, Scielo, and Science Direct. SciELO comprises the production of articles produced in several countries in Latin America; Wiley Online Library Covers international literature in the areas of Science and Information Technology since the mid-1960s; SCOPUS comprises several areas of knowledge, including history, education, psychology, linguistics and literature; and ScienceDirect is a research tool published by Elsevier that offers information for researchers, teachers, students, and health professionals.

After the search, a total of 125 publications were found using the terms: "Reconstituted corn grain silage" or "Rehydrated corn grain silage" or "Silagem de grãos de milho reidratados" or "Silagem de grãos de milho reconstituídos". Only articles that met the predetermined inclusion criteria were included in the systematic review.

For inclusion, studies needed to have the following standardized criteria: (1) one of the treatments did not include any additives in the silage (silage quality dataset) and if it evaluated silage in ruminant feed; and (2) included treatments comprising only beef cattle, goats, and sheep.

### 2.2. Data Mining

After conducting the search of publications in the databases, the articles were forwarded to the bibliographic reference management software Mendeley® (version 1.19.8 Installers, New York, NY, USA), which helped in the elimination of duplicate articles and in the organization of the abstracts. No studies were identified evaluating rehydrated corn silage as feed for beef cattle, goats, and sheep. After the refinement process, the remaining 20 articles were tabulated in an Excel® file to be evaluated in the screening process accord-

ing to the following information: (a) Author, (b) Journal, (c) Year, (d) Title, (e) Meets the selection criteria, (f) Does not meet the selection criteria, and (g) Reason why the article does not meet the selection criteria.

Based on the inclusion criteria, fifteen (15) peer-reviewed publications were sorted by first author, publication reference, additive used, number of replications, and standard error of the mean (SEM), and the following variables were extracted from rehydrated corn grain silage on the control (silage without additives) and treated (silage with additives) response, dry matter (DM), crude protein (CP), neutral detergent fiber (NDF), acid detergent fiber (ADF) concentrations, pH, lactic acid, acetate, propionate, butyrate, ethanol, counts of lactic acid bacteria (LAB), yeast, molds (CFU/g DM), effluent losses, gas losses, DM recovery, aerobic stability (h), and DM in vitro digestibility.

A flow chart explaining the study identification and selection process to analyze the effects of additives on the quality of rehydrated corn grain silage is shown in Figure 1.

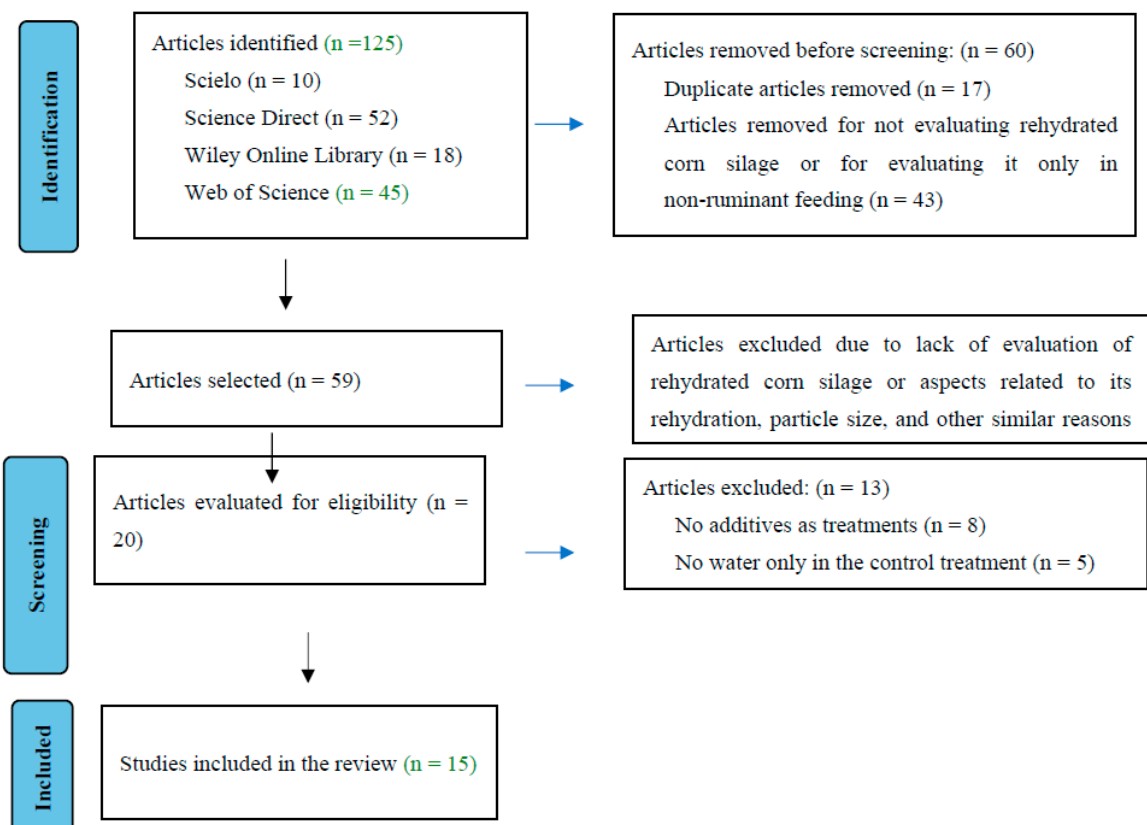

**Figure 1.** Flowchart showing the inclusion criteria for the selection of studies used to conduct the systematic review on the effects of additives on the quality of rehydrated corn grain silage.

### 2.3. Statistical Analysis

Data were analyzed using the SAS® statistical software (Statistical Analysis System, version 9.4). The results of the parameters evaluated were compared using analysis of variance (ANOVA) with a significance level of 5%. The model included a random effect of paired comparison within the study and a fixed effect of the treatments without additives and with additives. The mean, standard error of the mean, and minimum and maximum values were calculated for each treatment through the MEANS procedure.

## 3. Results

The study was carried out based on a sample of fifteen articles published between the years 2014 and 2023, as shown in Table 1. The found studies addressed the effects of additives on chemical composition, fermentation losses, and aerobic stability of silage. Two of the studies evaluated dry matter in vitro digestibility.

**Table 1.** Articles selected from the databases.

| ID | Reference | Title | Evaluated Parameters |
|----|-----------|-------|----------------------|
| 1 | Ferraretto et al. [41] | Effect of ensiling time on fermentation profile and ruminal in vitro starch digestibility in rehydrated corn with or without varied concentrations of wet brewers grains | Chemical composition |
| 2 | Rezende et al. [42] | Rehydration of corn grain with acid whey improves the silage quality | Chemical composition, fermentation parameters, and aerobic stability |
| 3 | Silva et al. [43] | Fermentation and aerobic stability of rehydrated corn grain silage treated with different doses of *Lactobacillus buchneri* or a combination of *Lactobacillus plantarum* and *Pediococcus acidilactici* | Chemical composition, fermentation parameters, and aerobic stability |
| 4 | Souza et al. [44] | Effect of rehydration with whey and inoculation with *Lactobacillus plantarum* and *Propionibacterium acidipropionici* on the chemical composition, microbiological dynamics, and fermentative losses of corn grain silage | Chemical composition and fermentation parameters |
| 5 | Cruz et al. [45] | Fermentative losses and chemical composition and in vitro digestibility of corn grain silage rehydrated with water or acid whey combined with bacterial-enzymatic inoculant | Chemical composition, fermentation parameters, and DM in vitro digestibility |
| 6 | Menezes et al. [46] | Effects of different moist orange pulp inclusions in the corn grain rehydration for silage production on chemical composition, fermentation, aerobic stability, microbiological profile, and losses | Chemical composition, fermentation parameters, aerobic stability, and DM in vitro digestibility |
| 7 | Jungues et al. [47] | Short communication: Influence of various proteolytic sources during fermentation of reconstituted corn grain silages | Chemical composition and fermentation parameters |
| 8 | Wang et al. [48] | Nutritional evaluation of wet brewers' grains as substitute for common vetch in ensiled total mixed ration. | Fermentation quality, nutritional value, aerobic stability, and in vitro gas production kinetics and digestibility |
| 9 | Wang et al. [49] | Fermentation quality, aerobic stability and in vitro gas production kinetics and digestibility in total mixed ration silage treated with lactic acid bacteria inoculants and antimicrobial additives. | Chemical composition, fermentative parameters, and aerobic stability |
| 10 | Carvalho et al. [50] | Fermentation profile and identification of lactic acid bacteria and yeasts of rehydrated corn kernel silage. | Chemical and microbiological characteristics |
| 11 | Pereira et al. [51] | Effect of cactus pear as a moistening additive in the production of rehydrated corn grain silage. | Fermentative and microbiological characteristics, aerobic stability, and chemical composition |
| 12 | Oliveira et al. [52] | Effect of Length of Storage and Chemical Additives on the Nutritive Value and Starch Degradability of Reconstituted Corn Grain Silage | Chemical composition, fermentation characteristics, and ruminal in situ degradability |

**Table 1.** *Cont.*

| ID | Reference | Title | Evaluated Parameters |
|---|---|---|---|
| 13 | Roseira et al. [53] | Effects of exogenous protease addition on fermentation and nutritive value of rehydrated corn and sorghum grains silages. | Fermentation and nutritive value |
| 14 | Silva Neto et al. [54] | Propionic acid-based additive with surfactant action on the feeding value of rehydrated corn grain silage for dairy cows performance. | Nutritional value and animal performance |
| 15 | Costa et al. [55] | Particle size and storage length affect fermentation and ruminal degradation of rehydrated corn grain silage | Chemical and microbiological characteristics, aerobic stability |

The articles were published in eight different scientific journals, namely, Journal of Dairy Science (three articles), *Semina: Ciencias Agrarias* (two articles), Animal Feed Science and Technology (two article), Animal Science Journal (three article), Agronomy (two article), Scientific reports (one article), Journal of Applied Microbiology (one article), and Livestock Science (one article). It is important to note that some additives in this review may fall into more than one category, as shown in Table 2.

**Table 2.** Description of additives added to rehydrated corn grain silages.

| Category | Additives | Classification [1] |
|---|---|---|
| By-product | Wet orange pulp | Fermentation stimulants/nutrients |
| | Wet brewery waste | Nutrients |
| By-product | Milk whey | Fermentation stimulants/nutrients |
| Bacterial inoculant | *Lactobacillus plantarum* | Fermentation stimulants |
| | *Pediococcus* | Fermentation stimulants |
| | *Lactobacillus buchneri* | Fermentation stimulants/Aerobic spoilage inhibitors |
| | *Enterococcus faecium* | Fermentation stimulants |
| | *Pediococcus acidilactici* | Fermentation stimulants |
| | *Propionibacteriu acidipropionici* | Fermentation stimulants |
| Enzymatic Inoculant | Cellulase and hemi-cellulase | Fermentation stimulants |
| Antimycotic agent | Natamycin | Fermentation inhibitors |
| Irradiation | Gama irradiation | Fermentation inhibitors |
| Chemical compound | Lactic acid | Fermentation stimulants |
| | Acetic acid | Fermentation inhibitors/Aerobic spoilage inhibitors |
| | Ethanol | Fermentation inhibitors |

[1] McDonald et al. [28].

Using the additives did not influence the chemical composition of the silage ($p > 0.05$) (Table 3). Regarding dry matter (DM) content, a value of 658 g kg$^{-1}$ was found in silages without additive and 644 g kg$^{-1}$ in silages with additive. It is worth noting that the fermentative process of these silages can affect the protein (CP) and neutral detergent fiber (NDF) content. However, although not significant in this review, some additives can influence these processes more.

**Table 3.** Chemical composition of rehydrated corn grain silage with and without additives.

| Item | | Rehydrated Corn Grain Silage | | $n$ [1] | SEM [2] | *p*-Value [3] |
|---|---|---|---|---|---|---|
| | | Without Additive | With Additive | | | |
| Chemical composition (g kg$^{-1}$ DM) | | | | | | |
| Dry matter | Mean | 658 | 644 | 39 | 8.91 | 0.27 |
| | Minimum | 586 | 564 | | | |
| | Maximum | 700 | 695 | | | |
| Crude protein | Mean | 90.3 | 93.0 | 39 | 3.67 | 0.48 |
| | Minimum | 70.0 | 74.7 | | | |
| | Maximum | 101 | 118 | | | |
| Neutral detergent fiber | Mean | 120 | 130 | 28 | 19.8 | 0.66 |
| | Minimum | 61.2 | 56.9 | | | |
| | Maximum | 214 | 232 | | | |
| Acid detergent fiber | Mean | 27.2 | 28.3 | 20 | 4.81 | 0.88 |
| | Minimum | 11.4 | 5.4 | | | |
| | Maximum | 38.2 | 62.2 | | | |

[1] $n$ = number of observations. [2] Standard error of the mean. [3] *p*-values significant at $p < 0.05$. Means followed by the same letter in the row are not statistically different.

As a result, they observed that bacterial activity was the main contributor to proteolysis (60%), followed by corn grain enzymes (30%), whereas fungi and fermentation end products (organic acids) had only minor contributions (~5% each) during the fermentation of rehydrated corn grain silage.

The pH values, organic acids, and ethanol concentration, as well as microbial population counts of the rehydrated corn grain silage without and with additives, are presented in Table 4.

**Table 4.** pH, organic acids, ethanol, and microbiology of rehydrated corn grain silage with and without additives.

| Item | | Rehydrated Corn Grain Silage | | $n$ [1] | SEM [2] | *p*-Value [3] |
|---|---|---|---|---|---|---|
| | | Without Additive | With Additive | | | |
| pH | Mean | 4.09 | 4.25 | 28 | 0.20 | 0.42 |
| | Minimum | 3.74 | 3.67 | | | |
| | Maximum | 4.94 | 5.66 | | | |
| Organic acids and ethanol (g kg$^{-1}$ DM) | | | | | | |
| Lactic acid | Mean | 15.4 | 15.54 | 27 | 2.21 | 0.99 |
| | Minimum | 9.07 | 0.90 | | | |
| | Maximum | 27.6 | 28.1 | | | |
| Acetic acid | Mean | 2.27 | 4.47 | 27 | 1.74 | 0.26 |
| | Minimum | 1.49 | 1.10 | | | |
| | Maximum | 3.60 | 16.2 | | | |
| Propionic acid | Mean | 0.54 | 0.68 | 23 | 0.38 | 0.62 |
| | Minimum | 0.03 | 0.01 | | | |
| | Maximum | 1.10 | 1.51 | | | |
| Butyric acid | Mean | 0.47 | 0.01 | 15 | 0.16 | 0.074 |
| | Minimum | 0.01 | 0.00 | | | |
| | Maximum | 1.71 | 0.14 | | | |
| Ethanol | Mean | 6.53 | 5.57 | 15 | 2.08 | 0.66 |
| | Minimum | 5.25 | 0.30 | | | |
| | Maximum | 7.16 | 12.5 | | | |

**Table 4.** *Cont.*

| Item | | Rehydrated Corn Grain Silage | | $n$ [1] | SEM [2] | *p*-Value [3] |
|---|---|---|---|---|---|---|
| | | **Without Additive** | **With Additive** | | | |
| | | Microbial population (log cfu g$^{-1}$) | | | | |
| Lactic acid bacteria (LAB) | Mean | 5.03 | 4.74 | 12 | 0.95 | 0.50 |
| | Minimum | 3.70 | 2.00 | | | |
| | Maximum | 6.10 | 6.28 | | | |
| Yeasts | Mean | 3.49 | 2.47 | 8 | 0.40 | 0.13 |
| | Minimum | 4.02 | 2.00 | | | |
| | Maximum | 4.23 | 3.37 | | | |
| Molds | Mean | 3.54 | 3.23 | 12 | 0.67 | 0.65 |
| | Minimum | 2.39 | 3.00 | | | |
| | Maximum | 4.51 | 4.85 | | | |

[1] $n$ = number of observations. [2] Standard error of the mean. [3] *p*-values significant at $p < 0.05$. Means followed by the same letter in the row are not statistically different.

In addition, fermentative losses, and aerobic stability (AS) of the silages were not influenced by the use or non-use of the additives ($p > 0.05$) (Table 5). Fermentative losses are usually not affected or are reduced due to the low moisture content of rehydrated corn grain silage. The in vitro digestibility of DM (IVDDM) was not influenced using additives ($p > 0.05$).

**Table 5.** Fermentative losses, aerobic stability, and DM in vitro digestibility of rehydrated corn grain silage with and without additives.

| Item | | Rehydrated Corn Grain Silage | | $n$ [1] | SEM [2] | *p*-Value [3] |
|---|---|---|---|---|---|---|
| | | **without Additive** | **with Additive** | | | |
| Effluent losses (kg/t [4]) | Mean | 2.36 | 3.05 | 8 | 1.07 | 0.55 |
| | Minimum | 2.12 | 1.23 | | | |
| | Maximum | 2.33 | 5.70 | | | |
| Gas losses (%) | Mean | 5.84 | 5.13 | 15 | 3.70 | 0.93 |
| | Minimum | 1.11 | 1.31 | | | |
| | Maximum | 12.3 | 21.2 | | | |
| Dry matter recovery (g kg$^{-1}$) | Mean | 965 | 976 | 25 | 7.48 | 0.14 |
| | Minimum | 941 | 936 | | | |
| | Maximum | 987 | 999 | | | |
| Aerobic stability (hours) | Mean | 96.2 | 98.9 | 23 | 42.7 | 0.95 |
| | Minimum | 36.0 | 25.5 | | | |
| | Maximum | 213 | 288 | | | |
| DM in vitro digestibility (g kg$^{-1}$ DM) | Mean | 875 | 839 | 8 | 48.1 | 0.77 |
| | Minimum | 805 | 786 | | | |
| | Maximum | 911 | 909 | | | |

[1] $n$ = number of observations. [2] Standard error of the mean. [3] *p*-values significant at $p < 0.05$. Means followed by the same letter in the row are not statistically different. [4] Tons.

The dry matter digestibility of silage can vary according to the type of culture used, especially rehydrated corn silage and the inoculated microorganisms (Figure 2), with an average ranging from 655 to 811 (g/100 g of ingested).

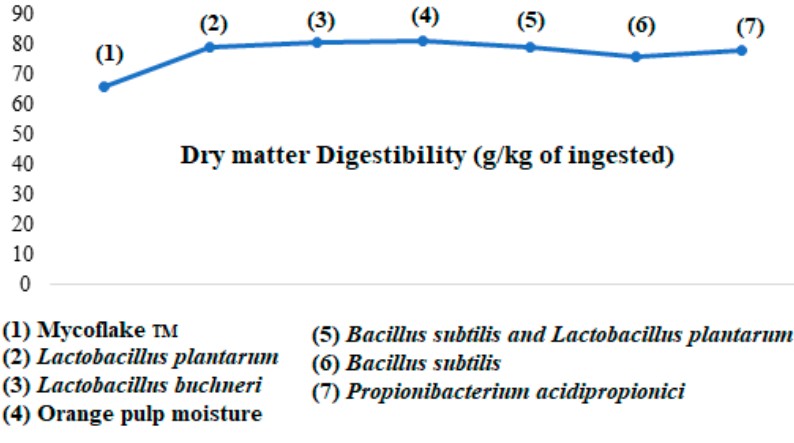

(1) **Mycoflake** ™
(2) *Lactobacillus plantarum*
(3) *Lactobacillus buchneri*
(4) **Orange pulp moisture**
(5) *Bacillus subtilis and Lactobacillus plantarum*
(6) *Bacillus subtilis*
(7) *Propionibacterium acidipropionici*

**Additives/Inoculant**

**Figure 2.** Dry Matter Digestibility (DMD) of corn silages retreated using microbial inoculants. Adapted from: Cruz et al. [45], Menezes et al. [46], Pereira et al. [51], Oliveira et al. [52]) and Lara et al. [56].

## 4. Discussion

From a total of the fifteen studies, thirteen were conducted in Brazil and two in the United States (Table 1). This higher number of studies developed in Brazil can be attributed to several factors, including the high cost of grain in recent years [57], and the traditional use of hard corn hybrids. These hybrids are usually used for their competitive agronomic characteristics in tropical conditions, such as resistance to insects in the field and during grain storage [41]. Due to these factors, rehydrated grain silage emerges as a viable alternative to increase starch digestibility, improving grain utilization efficiency in animal production [42].

In a study conducted in Brazil by Bernardes et al. [58], it was observed that 52.4% of milk producers incorporated the practice of using grain silage (corn or sorghum) in their animals' diets. A total of 16.6% corresponded to silage from rehydrated corn grains. This adoption gained prominence for several reasons. In particular, grain silage, such as corn, has proven beneficial in high-moisture situations [59,60], where conventional grain management can be challenging. This is due, in part, to greater flexibility in harvesting wet grains, which gives producers greater control over the harvesting process [61].

It is essential to highlight that particle size and storage time can affect the digestibility of rehydrated corn grain. Where prolonged storage time is directly related to improvements in starch digestibility, in contrast, there is an increase in dry matter loss. While particle sizes can influence the rate of degradation of prolamin in the silo, smaller particles increase the surface area available for microbial fixation, improving the extent of proteolysis in the silo and the hydrolysis of starch in the rumen. In contrast, when fine grain grinding is used, ensiling time increases, along with increased labor and energy expenditure [50,62].

In a study conducted by Jungs et al. [47] to evaluate the relative contribution of enzymes, bacteria, fungi, and fermentation products in the degradation of prolamins (zeins), it was observed that during the fermentation of corn grain silages with 79.2% vitreousness, bacteria played the predominant role in the degradation of the protein matrix involving starch granules, representing 60.4% of the proteolytic activity. Enzymes contributed 29.5%, whereas fungi and fermentation products contributed modestly, representing only 5.3% and 4.8%, respectively. These results indicate a clear differentiation in the contributions of these components to the degradation of prolamins during the fermentation process in corn silages.

However, the scarcity of technical information regarding the ensiling process leads producers to replace adequate management techniques with additives [63]. However, it is worth highlighting that it is essential to adopt practices during the ensiling process

to reduce the incidence of spoilage microorganisms [64]. According to Santos et al. [65], undesirable microorganisms are present during the ensiling process and are influenced by the type of plant and the presence of additives.

During the ensiling process, anaerobic fermentation produces organic compounds (lactic acids, acetic acids, and alcohol). These compounds can solubilize zein, making it more susceptible to subsequent enzymatic degradation and hydrolysis during digestion by rumen microorganisms. This solubilization contributes to releasing the starch associated with this protein, making it readily available for absorption in the animals' intestines [66].

The influence of cut length on plant material is a critical aspect of the ensiling process, directly and indirectly affecting the performance of animals that consume the silage. In silage production practice, it is crucial to distinguish cut length from other factors, such as the rupture of plant cells and the degree of post-silage compaction and sealing [67]. Obtaining precisely cut forage, stored in properly compacted and sealed silos, establishes a solid basis for effective anaerobic fermentation and minimization of losses [68].

Critical stages at which losses occur include field harvesting and silo filling, along with the silo respiration and fermentation phase. Furthermore, the production of effluents and exposure to oxygen during the storage phases [69]. Awareness of these stages is crucial to implementing effective management strategies to minimize losses throughout the ensuing process.

Losses during harvest are mainly associated with cutting and withering processes. Both steps result in the loss of non-structural carbohydrates due to plant respiration, cell rupture, and the leakage of cellular contents [70]. These losses include not only a decrease in the digestibility of the final product but also the possibility of contamination by mycotoxins, as observed by Jobim and Nussio [71].

Furthermore, cutting height plays a crucial role in losses due to soil contamination. Therefore, it is recommended to harvest forage crops in dry weather conditions, maintaining a minimum cutting height of 50 mm for grasses and 150 mm for corn crops, measured above ground level, as emphasized by Wilkinson [70].

The decrease in particle size and the disintegration of the cell wall structure contribute to the increase in food density, as observed by Pereira et al. [72]. Furthermore, the compaction process results in a lower oxygen concentration in the medium, reducing aerobiosis time, as highlighted by Neumann et al. [73]. Tan et al. [74] indicate that the effect of particle size is greater in materials with lower moisture content, exerting a more significant influence on the levels of ADF, NDF, hemicellulose, pH, and organic matter.

Drier forages present a challenge for compaction, allowing for a higher oxygen concentration in the silo. This, in turn, prolongs the plant's respiration time, reducing the soluble carbohydrate content and, consequently, impacting its nutritional value and lactic acid production [75].

The density of the material intended for silage harms losses during storage, compromising the maintenance and preservation of nutrients between the silo's opening and the silage's use period as food, as highlighted by Wilkinson and Davies [76]. Sucu et al. [77] emphasize that more efficient silage compaction contributes to better preserving soluble carbohydrates and proteins, resulting in lower losses of gases, effluents, and dry matter (DM). Silos with higher density present reduced levels of oxygen, creating ideal conditions for adequate fermentation from the moment the silo is closed, which results in lower effluent losses, as observed by Neumann et al. [73].

As Rinne and Seppälä [78] pointed out, the substrate availability for lactic acid bacteria and the effective removal of oxygen are essential characteristics for the rapid homofermentative fermentation of lactic acid in plant material. Packing density plays a significant role in gas flow within the ensiled mass, influencing air infiltration rates [75].

Lambs that received concentrate-based feed composed of 100% rehydrated corn grain silage, replacing dry grains, recorded a significant increase in weight gain, as distributed by [79]. This suggests that it can contribute substantially to a more efficient use of starch by ruminants. It is important to note that the rate of starch handling is influenced by the

ensiling process, as pointed out by [80,81]. McDonald et al. [28] classify silage additives into five main groups: fermentation stimulants, fermentation inhibitors, aerobic spoilage inhibitors, nutrients, and absorbents.

Rehydrated corn grain silage practice is explained by the low content of water-soluble carbohydrates in corn grains [23] and the reduced activity of lactic acid bacteria (LAB), caused by the limitation of water activity and the stress to which the grains are subjected during natural drying in the field. This can compromise the fermentation capacity of rehydrated corn grains [82]. The improvement of the silage quality using rehydrated corn grain silages is expected since it is common for these grains to be rehydrated until they reach 35 to 40% moisture to ensure good fermentation. Correct moisture content is necessary for the growth and reproduction of lactic acid bacteria, as mentioned by Hu et al. [83].

However, dry matter losses occur at all stages of the ensiling process, mainly due to cellular respiration during filling and sealing of the silo. The greatest fermentative losses result from the action of bacteria of the genus Clostridium. These microorganisms develop in moist silages and high pH (>5.0), producing mainly butyric acid, which is responsible for losses of up to 50% of dry matter and energy in the ensiled material [84].

During the fermentation of the ensiled mass, competition for substrates (soluble carbohydrates and lactate) occurs inside the silo, causing losses of dry matter and energy. However, the amount of these losses depends on the dominant microorganism of the fermentation and substrate used. Therefore, it is essential to use additives that promote lactic fermentation and reduce losses in ensiling [69].

LAB mainly includes *Lactobacillus*, *Pediococcus*, and *Streptococcus* species, which are found on the surface and internal structures of forage plants. When the plant is harvested and ensiled, these bacteria act, metabolizing the water-soluble sugars in plant cells. This metabolic process produces lactic acid as the main product, along with a smaller amount of acetic acid. These microorganisms positively influence the product's quality, resulting in improved health and, consequently, animal performance [58,85].

When the material is in anaerobiosis, heterolactic fermentation begins by bacteria of the genus *Lactococcus*, which has the function of acidifying the medium until the pH reaches values below 5.0, thereby inactivating these bacteria and favoring *Lactobacillus*, which become the dominant group in silage [86].

Silva et al. [87], when evaluating the stability of wet corn grain silages and rehydrated corn, we achieved a DM content of 65.5% during the analysis of rehydrated corn, which presented a moisture content of 35%. Faustino et al. [88] state that failure to mix vigorously when incorporating water into the corn can compromise grain hydration, potentially resulting in fungal growth and silage losses. The MS values found in this study surpassed the results of [59,89], a possible consequence of the physical grinding treatment of the material. Jobim et al. [90], when evaluating wet corn grain silage in feeding ruminants, point out that moisture contents above 35% when ensiling wet corn grains favor DM losses, which can alter the contents of nitrogen and soluble carbohydrates. These factors may influence treatments with 35%, 40%, and 45% water inclusion.

As for CP, it is possible that the use of additives changes the microbial profile of the silage, which would increase proteolysis of the hydrophobic protein matrix surrounding the starch granules, thus promoting greater starch digestibility [91]. Importantly, this increase in proteolysis can convert some of the true protein into soluble protein and $N-NH_3$. Organic acid-based chemical additives are the most used to control microbial growth [5]. Although only one study was included in this review evaluating chemical additives in silages, Junges et al. [47] used these additives with antifungal and fermentation inhibitory functions only to estimate the relative contribution of corn grain enzymes, bacteria, fungi, and fermentation end products to protein solubilization during fermentation.

Adding urea to roughage feeds increases protein content, improves roughage digestibility through changes in the fibrous fractions, and reduces losses associated with the fermentation process in silages [92]. Pádua et al. [93], evaluating the effects of urea

doses on potato grass hay (*Paspalum Notatum*), observed that the CP content increased with increasing levels of urea used in the study. Similarly, Sousa et al. [44], evaluating the chemical composition of corn straw ammoniated with urea added at 0, 2, 4, 6, and 8% of total DM, observed an increase in CP content with the inclusion of doses of urea with values ranging from 5.36 to 11.2 (%DM).

In addition, ensiling time is another factor influencing grain proteolysis [94]. According to Xu et al. [95], the intensity of proteolysis is higher at the beginning of the storage period due to the higher concentration of lactic acid bacteria. However, it is maintained until the opening of the silo. Wang et al. [48], evaluating the effect of replacing common vetch with wet brewers' grains at different proportions on the fermentation quality, the nutritive value of the ensiled total mixed ration, observed higher CP content (28.6% CP as basis DM) with wet brewers' grains addition. Ferraretto et al. [4] and Mambach et al. [59] observed that the use of microbial additives did not promote significant changes in the crude protein (CP) content of the ensiled material, corroborating with Sebastian et al. [96], evaluating the inoculation of wet corn grain silages (average of 7.31% CP). However, Schaefer et al. [97] recorded a drop in CP content in wet corn grain silages, whether inoculated or not, obtaining averages of 9.9 and 10.2% CP, respectively. These findings are relevant for developing strategies that allow an adequate degree of proteolysis, especially for corn grains with lower nutrient digestibility due to their high vitreousness or advanced maturity, without compromising the fermentative characteristics.

In contrast, the NDF content of silage may be reduced due to acid hydrolysis of hemicelluloses [98,99], which results in increased availability of soluble substrates. Although it is not possible to compare NDF content before and after ensiling in this review because there are no data on the chemical composition of the mass before ensiling, previous studies, such as that of Rezende et al. [42], indicate greater reductions in NDF in silage when whey is used to rehydrate corn grains when compared to water. However, Sousa et al. [44] and Cruz et al. [45] found different effects of whey compared to water, making it difficult to evaluate this additive's intrinsic effect in reducing the silage's NDF content.

During the proper fermentative process in rehydrated corn grain silages, lactic fermentation generates mainly lactic acid and, to a lesser extent, acetic acid from carbohydrates present [100,101]. As a result, the pH is reduced to a level where undesirable fermentation is prevented. The increased *Clostridium* population results in utilizing desired fermentation products such as lactic acid and sugars, proteins, and amino acids to form butyric acid and amine, amine, and ammonia [102].

Therefore, pH measurements, concentrations of organic acids, alcohols, N-NH$_3$, and quantification of microbial populations are most used to assess silage fermentation [45]. In this review, it was impossible to include the N-NH$_3$ concentration of silages since the data obtained in the studies presented different ways of quantification. However, it is important to note that N-NH$_3$ is an accurate indicator of proteolysis during grain ensiling [49,102]. Overall, silages showed adequate fermentation, shown mainly by the lower pH and higher lactic acid content.

Increasing soluble protein (CP) and NH$_3$-N concentrations over silage periods indicate continuous degradation of the protein matrix, even with prolonged storage times. These results corroborate with the findings of Hoffman et al. [91], who observed continuous degradation activity of the protein matrix up to 240 days of storage. However, it is interesting to note the discrepancy with the results of Oliveira et al. [52], who reported a different pattern for NH$_3$-N in silages treated with Mycoflake™, showing a slight increase in CP levels after 30 days of ensiling, indicating possibly an increase in matrix proteolysis; the concentration of NH$_3$-N was considerably higher, reaching similar levels. These observations highlight the complexity and variability in the effects of additives on the composition and degradation of the protein matrix in silages.

The mean pH of the silages is within or very close to the desirable standards for good fermentation, which should be between 3.8 and 4.2 [28]. The concentrations of lactic acid averaged 15.4 g kg$^{-1}$ DM. Wang et al. [48] stated that the pH of the treatments increased

during 14 days of aerobic exposure, indicating that the treatments remained stable. It was probably due to the sufficient content of acetic acid (11.5 g/kg DM) and propionic acid (2.74 g/kg DM) in the silage after exposure to air. Although differences were found in the classifications of the additives used in rehydrated corn grain silages, it is important to note that analysis of the effect of these additives on the silage microbiota is difficult to perform due to the scarcity of studies that have performed this analysis.

The diversity of microorganisms in silage is altered according to the characteristics of the forage crop, with a succession of genera and species as the conditions of the environment change [103]. Among these phyla are lactic acid and propionic acid bacteria, which are desirable during ensiling. In contrast, undesirable microorganisms, such as enterobacteria, molds, and yeasts, perform detrimental activities during the ensiling process because they compete with lactic acid bacteria in sugar fermentation [104]. Despite the different classifications of additives used in rehydrated corn grain silages, it has yet to be possible to evaluate the effect of these additives on the silage microbiota due to the small number of studies that have performed this analysis.

However, the AS can be improved when undesirable microorganisms are inhibited with appropriate additives after silo opening [105–107], as is the case with the bacterial inoculant *Lactobacillus buchneri* [23,108]. Silva et al. [23] evaluated the effects of the bacterial inoculants *Lactobacillus plantarum* and *Pediococcus* in different concentrations, $1.0 \times 10^5$, $5.0 \times 10^5$, and $1.0 \times 10^6$ CFU/g, and *Lactobacillus buchneri* in the same concentrations previously mentioned, on the silage of rehydrated corn grains ensiled for 124 days, and observed that the inoculant *Lactobacillus buchneri* at a dose of $1 \times 10^5$ CFU/g is a viable strategy to increase the aerobic stability of rehydrated corn grain silage. Furthermore, they observed that homolactic bacteria did not alter the fermentation process or decrease the aerobic stability of the silages. The increase in aerobic stability may also occur with increasing storage time due to the gradual accumulation of fermentation products with antifungal properties, such as acetic acid and propionic acid [23]. According to Wilkinson et al. [109], acetic acid can enter the cells of microorganisms in an undissociated form in a low-pH environment (<4.73). Then, undissociated acetic acid would inhibit the acid–base balance in the cells of microorganisms, which slow growth and potentially lead to cell death. In addition, propionic acid and butyric acid have also been proven to be effective inhibitors of aerobic spoilage [109]. Furthermore, it has been reported that certain antifungal compounds were produced when legumes were ensiled [110–112].

Acetic acid is a crucial component in silage fermentation thanks to its antifungal effects. Interestingly, although the degree of grinding impacted other fermentation products, such as lactic acid and ethanol, it did not affect the concentration of this acid. Different microorganisms, such as heterofermentative LAB, enterobacteria, Bacillus, and acetic bacteria, contribute to the production of acetic acid, as observed by Almeida Carvalho-Estrada et al. [64] and Fernandes et al. [113].

The lack of effect can be explained by the small number of studies that performed this analysis. It is important to note that, due to the fermentation process, the digestibility of these silages can be improved by the higher content of proteolysis in the grains; however, digestibility can be decreased by increasing the hydrolysis of hemicellulose, which decreases the content of NDF and consequently increases the content of ADF of the silage. The in vitro digestibility of DM (IVDDM) was influenced by additives ($p = 0.041$). Two studies evaluated the IVDDM of rehydrated corn grain silage (Table 5). Despite the small number of studies, each study will be evaluated separately to assess the influence of additives better.

The first study evaluated the inclusion of wet orange pulp, a byproduct of the orange processing industry, used due to its high pectin and soluble carbohydrate content [114]. Wet orange pulp was included in proportions of 21, 34, and 42% in corn grains' natural matter (as fed) and ensiled for 62 days [46]. The inclusion of orange pulp reduced the NDF tors due to acid hydrolysis during the storage process, solubilizing the hemicellulose and increasing the acid detergent fiber and lignin contents, leading to a reduction in the IVDDM of the silage with greater inclusion of orange pulp [115]. The second study evaluated whey

(until the grains reached 35% moisture) associated with the bacterial inoculants *Pediococcus acidilactici*, *Propionibacterium acidipropionici*, *Enterococcus faecium* ($1.0 \times 1010$ CFU/g), an enzyme complex, and cellulase and hemicellulase at 5%, ensiled for 60 days [116]. Whey was used as a source of moisture because it is rich in soluble carbohydrates (lactose), proteins, minerals, and vitamins, as well as a variable amount of lactic acid [79] and LAB [117]. However, the authors found no difference in IVDDM between the evaluated treatments. In other words, IVDDM was more affected by the wet orange pulp additive due to the more significant acid hydrolysis of hemicellulose provided by the higher NDF content of this additive than by the use of whey associated with bacterial and enzymatic inoculants [118].

However, rehydrated grain silage requires longer storage times to increase dry matter digestibility significantly [108,113]. The in situ digestibility of starch from rehydrated corn grain silage was observed by Fernandes et al. [113] after 120 days of fermentation, compared to corn in silage, with values of 92.0 and 72.0%, respectively.

As noted by Nair et al. [119] and Muck et al. [110] in studies involving whole plant silages with the combination of *Lactobacillus buchneri* and *Lactobacillus hilgardii*, or without the addition of inoculants, the use of inoculants had no significant impact on rumen fermentation, digestibility total nutrients in the gastrointestinal tract or performance of beef cattle. However, it was noted that cattle fed with silage containing the inoculants showed greater feed efficiency compared to those fed with control silage (without inoculant). This result suggests a potential benefit in feed efficiency associated with using these specific inoculants in silages.

When comparing the inclusion of rehydrated grains replacing dry grains in the diet of cattle in confinement, an improvement in IVDDM and protein degradability was observed, as reported by Berton et al. [120]. These results were consistent with Pereira et al. [121], who affirmed that the size of the grain affects IVDDM, as the finer crushed and rehydrated corn grain had greater digestibility (71.6%) than the coarser rehydrated corn grain (42.8%). The quality of the silage is determined by its ability to partially meet the animals' nutritional needs, influenced by animal intake and nutritional value (chemical composition) [122,123]. However, animal intake can be altered by constantly changing factors, consequently influencing the nutritional value [56,124–126].

DM digestibility is used as a parameter in evaluating the quality of foods, which may not only contain nutrients; however, they are available for use by microorganisms in the rumen (Graph 1). Lara et al. [56] found differences in the digestibility of corn silage DM using Bacillus subtilis and the combination of *B. subtilis* and *L. plantarum*, with the combined inoculant being the one with the highest digestibility. However, Cruz et al. [45], evaluating the enzymatic–bacterial inoculant used, was composed of *L. plantarum*, *L. buchneri*, and *L. lactis* and showed an improvement in DM digestibility. This improvement is associated with the breaking of the bonds between prolamins and starch and increasing the storage time of corn grain silages [46,51,68].

The silage storage period plays a crucial role in the degradation of the protein matrix. A minimum storage time of 60 days is recommended for rehydrated corn grain silages (RCGS), as the greatest increases in proteolysis and starch availability occur in the first 60 days after ensiling, as highlighted by Fernandes et al. [113]. Finally, a minimum storage period of 52 days is recommended for rehydrated corn grain silages to maximize the protein matrix breakdown effects and increase digestibility [23].

### 5. Conclusions

Using additives in corn silage is a promising practice that can significantly benefit silage fermentation. According to the literature, moisture silage additives are expected to directly inhibit clostridia and other detrimental microorganisms, mitigate high mycotoxin levels, enhance aerobic stability, improve cell wall digestibility, and increase the efficiency of utilization of silage nitrogen by ruminants.

The use of fermentation-stimulating additives (*Lactobacillus buchneri*), as described and presented in the studies conducted, can improve the quality of rehydrated corn grain silage. In addition, the effects of additives in rehydrated corn grain silages on animal diets demonstrate improved performance due to greater starch availability for ruminal energy production and lower enteric methane production. There are still a few studies and more research to elucidate the best additives and the ideal amount to be added to ground corn grain silages.

**Author Contributions:** Conceptualization, Data curation, Formal analyses, Investigation, Writing—Original draft, L.V.D.; methodology, J.M.P.F. and R.L.O.; investigation, P.H.S.M. and E.M.S.; writing—review and editing, R.L.E., J.P.F.d.O. and R.R.d.N.; visualization, E.J.S.A.; funding acquisition, L.R.B. All authors have read and agreed to the published version of the manuscript.

**Funding:** The research was financially supported by the National Council for Scientific and Technological Development (Brazil), with grant Number 441321/2017-8, and Research Support Foundation of the State of Paraiba with grant Number 3075/2021.

**Institutional Review Board Statement:** Not applicable.

**Informed Consent Statement:** Not applicable.

**Data Availability Statement:** Data sharing not applicable.

**Acknowledgments:** We have thanked the Federal University of Campina Grande through facilities support. All individuals included in this section have consented to the acknowledgment.

**Conflicts of Interest:** The authors declare no conflict of interest.

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
