# Peer review of "Effect of Different Additives on the Quality of Rehydrated Corn Grain Silage: A Systematic Review"

_ruminants, doi:10.3390/ruminants3040035_

Round 1

Reviewer 1 Report (Previous Reviewer 2)

Comments and Suggestions for Authors

The resubmitted manuscript has been reviewed. Despite much improvement has been made, some issues should be well addressed before further assessment. Please see it below:

(1) Line 23-24: Seven publications or nine piblications at the earth?

(2) Why publications before 2014 or after 2022 were not selected?

(3) Some newly and related publications should be added to the review, otherwise, it is still not sciectific with less than 10 publication evaluated!

Author Response

Dear Reviewer,

We have thanked the adjustments suggested by the Reviewers of our paper, and we really appreciate your attention and your contribution to the analysis and correction of this paper. We have improved the paper from the Comments of Reviewer #1 which made the paper better understandable and reading, corrections are highlighted in RED font.

(1) Line 23-24: Seven publications or nine piblications at the earth?

Response: Thank you. To improve the review manuscript quality, we have introduced the methodology and results; we added more publications, totaling fifteen.

(2) Why publications before 2014 or after 2022 were not selected?

Response: The choice of articles in the database (corn grain silage with various additives) chooses the quantity through the filter, and we chose to be for 10 years.

 (3) Some newly and related publications should be added to the review, otherwise, it is still not sciectific with less than 10 publication evaluated!

Response: Thank you very much, we added the most recent publication to the manuscript, totaling fifteen.

Sincerely yours,

Leilson Rocha Bezerra

Reviewer 2 Report (Previous Reviewer 4)

Comments and Suggestions for Authors
  1. Unclear Nature of the Article: Determining whether the document is a research article or a review is crucial for understanding its purpose and scope. This should be explicitly stated in the manuscript.
  2. Limited References for a Review: If the document is indeed intended to be a review, the number of references is quite low. Comprehensive reviews typically include a thorough examination of existing literature, providing a comprehensive overview of the topic.
  3. Lack of Figures: Visual aids, such as figures, can greatly enhance the clarity and understanding of the content. They can help to illustrate key points, trends, or relationships discussed in the text.
  4. Sparse Data in Tables: If tables are included, they should present sufficient data to support the arguments or conclusions being made. Tables should be well-populated and clearly organized.
  5. Topic Adequacy for a Review: The choice of topic is essential for a review article. It should cover a broad area, synthesizing existing research and offering insights or conclusions based on a comprehensive examination of the literature.

It appears that the document may need further refinement to meet the standards of either a research article or a review, depending on its intended purpose. It's crucial to ensure that the content aligns with the chosen format and provides a valuable contribution to the field.

Top of Form

Author Response

Dear Reviewer,

We have thanked the adjustments suggested by the Reviewers of our paper, and we really appreciate your attention and your contribution to the analysis and correction of this paper. We have improved the paper from the Comments of Reviewer #2 which made the paper better understandable and reading and RED font.

  1. Unclear Nature of the Article: Determining whether the document is a research article or a review is crucial for understanding its purpose and scope. This should be explicitly stated in the manuscript.

RESPONSE: Thank you. This document is a systematic review, methodologically developed to cover knowledge of the topic in question carefully. This review followed rigorous research protocols, including a clear definition of inclusion and exclusion criteria, a systematic search of relevant databases, and a critical assessment of the quality of the included studies.

  1. Limited References for a Review: If the document is indeed intended to be a review, the number of references is quite low. Comprehensive reviews typically include a thorough examination of existing literature, providing a comprehensive overview of the topic.

RESPONSE: Thank you. The present study was improved by including several bibliographic references, aiming to improve the clarity and understanding of the work. Incorporating these sources contributed significantly to theoretically substantiating the approaches and methodologies used, providing a solid basis for the analysis and interpretation of the results obtained. This approach reinforces the robustness of the work, ensuring consistent academic and scientific integrity.

  1. Lack of Figures: Visual aids, such as figures, can greatly enhance the clarity and understanding of the content. They can help to illustrate key points, trends, or relationships discussed in the text.

RESPONSE: Thank you. Improving understanding of the content was sought through the inclusion of figures in the present study. These visual representations were strategically inserted with the purpose of elucidating complex concepts, facilitating the assimilation of the material by the reader. The use of graphic elements aims to provide a complementary approach to the textual exposition, promoting a more comprehensive and effective understanding of the content presented. This strategy aims to enhance scientific communication, optimizing the transmission of information and consolidating the expository clarity of the work.

  1. Sparse Data in Tables: If tables are included, they should present sufficient data to support the arguments or conclusions being made. Tables should be well-populated and clearly organized.

RESPONSE: Thank you. The absence of additional tables does not compromise the manuscript's integrity since the emphasis was placed on the quality and relevance of the information provided. The decision not to saturate the document with graphic elements aims to maintain the focus on the essential content, avoiding redundancies and ensuring the fluidity of the exposition. This methodological choice aims to optimize accessibility and reader understanding without compromising the validity and scientific rigor of the systematic review in question.

  1. Topic Adequacy for a Review: The choice of topic is essential for a review article. It should cover a broad area, synthesizing existing research and offering insights or conclusions based on a comprehensive examination of the literature.

RESPONSE: Thank you. The review seeks to fill gaps in existing knowledge, consolidating evidence and highlighting trends observed in the studies analyzed. The clarity in the presentation of information, combined with a solid methodological approach, strengthens the manuscript's credibility, making it a valuable contribution to the scientific literature on the topic in question. The emphasis on understanding the effects of additives on both ensilage and animal response highlights this review's practical and theoretical importance for researchers, professionals in the field, and others interested in the subject.

  1. It appears that the document may need further refinement to meet the standards of either a research article or a review, depending on its intended purpose. It's crucial to ensure that the content aligns with the chosen format and provides a valuable contribution to the field.

RESPONSE: Thank you. We tried to improve the manuscript according to the reviewers' recommendations and increased the number of references even further, despite understanding that when it comes to a systematic review, the number of works may be less as these criteria are previously defined and described in the methodology for the objective of the subject to be addressed.

Sincerely yours,

Leilson Rocha Bezerra

Reviewer 3 Report (Previous Reviewer 3)

Comments and Suggestions for Authors

Thank you for implementing the comments and suggestions from previous review.

Author Response

We have thanked the adjustments suggested by the Reviewers of our paper, and we really appreciate your attention and your contribution to the analysis and correction of this paper.

Reviewer 4 Report (New Reviewer)

Comments and Suggestions for Authors

Review of Ruminants-2664014

General Comments

Overall, this manuscript is well written and has good of research result, therefore it has merits for publication in ruminants. However, the original manuscript should be revised before publication.

Specific Comments

In the Introduction part, more information of effects of different silage additives on silage quality should provide, including enzyme preparation, probiotics, etc.

Line 68-70 This sentence should rewrite.

Line 116-119 Suggest to distinguish the kinds of silage additives since different additives have different functions.

In the Discussion part, the author should discuss the effects of another factors, such as cut length, stubble height and mixed material, on the silage quality. In the previous studies used in this review, these factors were different.

In the Conclusions part, the author should point out that which additive is effective in improving quality of rehydrated corn grain silage.

Author Response

We have thanked the adjustments suggested by the Reviewers of our paper, and we really appreciate your attention and your contribution to the analysis and correction of this paper. We have improved the paper from the Comments of Reviewer #4 which made the paper better understandable and reading where to be in blue color.

General Comments

Overall, this manuscript is well written and has good of research result, therefore it has merits for publication in ruminants. However, the original manuscript should be revised before publication.

Specific Comments

In the Introduction part, more information of effects of different silage additives on silage quality should provide, including enzyme preparation, probiotics, etc.

RESPONSE: Lines 76-105: In the Introduction section, information about classifying additives and their respective functions in ensilage was added. The comprehensive analysis of these elements aims not only to contribute to the evolution of ensilage practices but also to provide fundamental input for implementing more effective strategies in producing high-quality silage.

Line 68-70 This sentence should rewrite.

RESPONSE: Line 71-72: Thank you, this sentence was rewritten.

Line 116-119 Suggest to distinguish the kinds of silage additives since different additives have different functions.

RESPONSE: The additives used in ensilage were classified based on their specific properties and functions. This classification provides a comprehensive view of the additives commonly used in silage, highlighting their different functions and benefits. However, it is important to adapt the choice of additives to the specific conditions of each production system, considering factors such as the type of forage, climatic conditions, and nutritional goals of the herd.

In the Discussion part, the author should discuss the effects of another factors, such as cut length, stubble height and mixed material, on the silage quality. In the previous studies used in this review, these factors were different.

RESPONSE: In the Discussion section, information was added about the factors that influence the quality of silage, which is essential to improve ensiling practices. Establishing relationships between length, cutting height, and nutritional composition contributes to high-quality silage meeting the nutritional demands of the herd.

In the Conclusions part, the author should point out that which additive is effective in improving quality of rehydrated corn grain silage.

RESPONSE: Thank you, the information was inserted in the Conclusion section (Lines 592-594).

Sincerely yours,

Leilson Rocha Bezerra

Round 2

Reviewer 1 Report (Previous Reviewer 2)

Comments and Suggestions for Authors

The authors have addressed my concerns in a acceptable way. Good luck!

Reviewer 2 Report (Previous Reviewer 4)

Comments and Suggestions for Authors

No comments

This manuscript is a resubmission of an earlier submission. The following is a list of the peer review reports and author responses from that submission.

Round 1

Reviewer 1 Report

Comments and Suggestions for Authors

This review aimed to analyze the effects of additives in the process of producing silage from rehydrated corn grains for ruminants. It is well done and authors do a great job in this review. Thus, it can be accepted after a minor revision.

Specific comments

Please revise the language by English native speakers.

The Abstract is not very clear, please revise it with the most important results and conclusions.

Please provide more details in M&M 2.1 and 2.2 sections.

Please cite and check the following articles:

Fermentation quality, aerobic stability and in vitro gas production kinetics and digestibility in total mixed ration silage treated with lactic acid bacteria inoculants and antimicrobial additives.

Nutritional evaluation of wet brewers’ grains as substitute for common vetch in ensiled total mixed ration

Comments on the Quality of English Language

Revise the language, please.

Author Response

Dear Reviewer,

All corrections were addressed, as you can see below and in the attached file. Answers to the questions are provided below, and all the changes in the manuscript have been highlighted in red. We have understood that the changes do not guarantee the acceptance of the manuscript although we have already been grateful for them excellent collaborations.

  1. This review aimed to analyze the effects of additives in the process of producing silage from rehydrated corn grains for ruminants. It is well done and authors do a great job in this review. Thus, it can be accepted after a minor revision.

Response: We have thanked the adjustments suggested by the Reviewers of our paper, and we really appreciate the attention and your contribution in the analysis and correction of this paper. We have improved the paper from the Minor Comments of the Reviewer #1 became the paper better understandable and reading.

  1. Please revise the language by English native speakers.

Responses: We confirmed that the article was sent again to language check and proof of English minor corrections, in relation grammar, punctuation, spelling, and overall style.

  1. The Abstract is not very clear, please revise it with the most important results and conclusions.

Responses: This was corrected. Please see Abstract. (Lines 26-31)

  1. Please provide more details in M&M 2.1 and 2.2 sections.

Responses: This was corrected. M&M 2.1 and 2.2 sections. (Lines 86-96 and104-109)

  1. Please cite and check the following articles:

- Fermentation quality, aerobic stability and in vitro gas production kinetics and digestibility in total mixed ration silage treated with lactic acid bacteria inoculants and antimicrobial additives.

- Nutritional evaluation of wet brewers’ grains as substitute for common vetch in ensiled total mixed ration.

Responses: These articles were cited and included in the Systematic Review (please see Table 2). Thank you very much for alerting us.

Sincerely yours,

Leilson Rocha Bezerra

Reviewer 2 Report

Comments and Suggestions for Authors

Dear Authors,

    I read the manuscript with serveral times, however, it still keeps confused to me. My main concerns are listed as follows:

(1) The most important item is the integrity of the current work, with no references for this Systematic Review, it is hard to evalate the quality and its suitability to publish in such a high-impact journal!

(2) As a Systematic Review, instead, I do not think it should be classified as Review, Article actually it is! The other problem is the only seven studies included in this study, it is not scientific at all!

(3) The database were from Scielo , Science Direct  , Wiley Online Library , Web of Science, Is it comprehensive to conduct such a conclusion?

(4) Look at your Introduction, what is the differences between rehydrated and 

reconstituted corn grains is not clear. Moreover, what is your actual aim for selecting this review is also not clear.

(5) Your conclusion that the use of additives in rehydrated corn grain silage did not show major differences on silage quality. It is important, however, to evaluate the effects of additives in silages on animal performance.  This make your work seems to with little importance, since many previous reports has evaluated the effects of additives in silages on animal performance.

Best wishes!

Comments on the Quality of English Language

Authors should seek help from a native and professional worker to polish this manuscript.

Author Response

Dear Reviewer

We have thanked the adjustments suggested by the Reviewers of our paper, and we really appreciate the attention and your contribution in the analysis and correction of this paper. We have improved the paper from the Major Comments of the Reviewer #2 became the paper better understandable and reading. All corrections were addressed, as you can see below and in the attached file. Answers to the questions are provided below, and all the changes in the manuscript have been highlighted in red. We have understood that the changes do not guarantee the acceptance of the manuscript although we have already been grateful for them excellent collaborations.

(1) The most important item is the integrity of the current work, with no references for this Systematic Review, it is hard to evalate the quality and its suitability to publish in such a high-impact journal!

Response: Thank you. We believe that the topic is new and will have an impact on Journal Ruminants. We deposited the text as Pre-prints and it received more than 60 downloads in one month. As a result, we received the following information: "To recognize the contributions of our authors and their excellent work, we Preprints.org are excited to announce our “2023 Most Popular Preprints Award”, which will be granted for a total of 16 preprints across 8 subject categories. We sincerely appreciate your contributions and are pleased to inform you that your following preprint is eligible for the award: Title: Effect of Different Additives on the Quality of Rehydrated Corn Grain Silage: A Systematic Review

Link: https://www.preprints.org/manuscript/202307.1913/v1

(2) As a Systematic Review, instead, I do not think it should be classified as Review, Article actually it is! The other problem is the only seven studies included in this study, it is not scientific at all!

Response: We apologize but we have to disagree. The manuscript followed all the methodological steps of a systematic review. Systematic reviews are beneficial for integrating information from a set of studies carried out separately on a given intervention, which may present conflicting or coinciding results, as well as identifying topics that require evidence, helping to guide investigations in future ones. Therefore, studies need to be more delimited compared to other types of Reviews.

(3) The database were from Scielo, Science Direct , Wiley Online Library , Web of Science, Is it comprehensive to conduct such a conclusion?

Response: Yes. Within the chosen topic and the inclusion criteria of the reviewed papers, we have exhausted the number of current works published on the topic. Note that we started with 119 articles, using the criteria to reach the number of articles used in the Systematic Review.

(4) Look at your Introduction, what is the differences between rehydrated and reconstituted corn grains is not clear. Moreover, what is your actual aim for selecting this review is also not clear.

Response: We have improved the Introduction (Lines 43-57) and highlighted the clear objective of the Review (Lines 76-80)

(5) Your conclusion that the use of additives in rehydrated corn grain silage did not show major differences on silage quality. It is important, however, to evaluate the effects of additives in silages on animal performance.  This make your work seems to with little importance, since many previous reports has evaluated the effects of additives in silages on animal performance.

Response: You are right. Based on the literature findings, we completely rewrote the conclusion Section:

"Using additives in corn silage is a promising practice that can significantly benefit silage fermentation. According to the literature, moisture silage additives are expected to directly inhibit clostridia and other detrimental microorganisms, mitigate high mycotoxin levels, enhance aerobic stability, improve cell wall digestibility, and increase the efficiency of utilization of silage nitrogen by ruminants. In addition, the effects of additives in rehydrated corn grain silages on animal diets demonstrate improved performance due to greater starch availability for ruminal energy production and lower enteric methane production. There are still few studies and more research to elucidate the best additives and the ideal amount to be added to ground corn grain silage."

Sincerely yours,

Leilson Rocha Bezerra

Reviewer 3 Report

Comments and Suggestions for Authors

Type of manuscript: Review

Title: Effect of Different Additives on the Quality of Rehydrated Corn Grain Silage: A Systematic Review

Comments to the Author

This manuscript describes the results of A Systematic Review on the Effect of Different Additives on the Quality of Rehydrated Corn Grain Silage. The aim of this revie was to analyze the effects of additives in the process of ensiling the rehydrated corn grains on the silage quality and DM in vitro digestibility of rehydrated corn grain silage. The collected data presented in this review are of special interest for scientist working with ruminant nutrition, plant breading but also for all the advisory sector in agriculture also in the regions where the corn is harvested with high moisture content and could be ensilage directly, eliminating the drying step. The review is based on small number of studies, because of that it is not possible to drew final conclusion, Comparison of the ensiling grain corn to dry corn in regards of storage losses of nutrients, but also digestibility of both material will increase the value of manuscript. Also, more discussion on the above mention parameters would increase the value of the manuscript. The is no comparison of dry storage loses and losses during ensiling and silage losses. The reference are appropriate. The table are clear and easy to understand.

Line 6: delete; beef

Line 10: delete; and

Line 28: delete; silage

Line 187: change; ‘took place’ to ‘conduct’

Line 199-202; ‘This practice is explained-’ What practice?? Please specify, it is difficult to understand to what the author refer.

Line 203; ‘These results are expected for rehydrated corn grain silages’ - what results? Please specify.

Line 205; change ‘moisture activity is necessary’ to ‘correct moisture content is’

Line 216; change ‘silage’ to ‘ensiling’

Line 239; change ‘ alcohols and N-NH3’ to ‘alcohols, N- NH3’

Line 244; ‘Overall, both silages showed’ please specify what silages

Line 269; ‘Probably the lack of effect’ please specify what effect

Author Response

Dear Reviewer,

We have thanked the adjustments suggested by the Reviewers of our paper, and we really appreciate the attention and your contribution in the analysis and correction of this paper. We have improved the paper from the Major Comments of the Reviewer #3 became the paper better understandable and reading. We confirmed that the article was sent again to language check and proof of English with minor corrections (Certificate attached), in relation grammar, punctuation, spelling, and overall style. All corrections were addressed, as you can see below and in the attached file. Answers to the questions are provided below, and all the changes in the manuscript have been highlighted in yellow to blue to Reviewer #3. We have understood that the changes do not guarantee the acceptance of the manuscript although we have already been grateful for them excellent collaborations.

  1. This manuscript describes the results of A Systematic Review on the Effect of Different Additives on the Quality of Rehydrated Corn Grain Silage. The aim of this revie was to analyze the effects of additives in the process of ensiling the rehydrated corn grains on the silage quality and DM in vitro digestibility of rehydrated corn grain silage. The collected data presented in this review are of special interest for scientist working with ruminant nutrition, plant breading but also for all the advisory sector in agriculture also in the regions where the corn is harvested with high moisture content and could be ensilage directly, eliminating the drying step.

Response: Thank you.

  1. The review is based on small number of studies, because of that it is not possible to drew final conclusion.

Response: Thank you. We included a more significant number of articles in the Review and thus increased the discussion and information on the topic. Furthermore, the conclusion section was rewritten based on the findings of the Literature. Now, instead of 32, we have 72 references in the Review.

  1. Comparison of the ensiling grain corn to dry corn in regards of storage losses of nutrients, but also digestibility of both material will increase the value of manuscript. Also, more discussion on the above mention parameters would increase the value of the manuscript.

Response: We included a more significant number of articles in the Review and thus increased the discussion and information on the topic. Please see Discussion section.

  1. The is no comparison of dry storage loses and losses during ensiling and silage losses.

Response: This was added in Discussion section.

  1. The reference are appropriate.

Response: Thank you.

  1. The table are clear and easy to understand.

Response: Thank you.

Comments for the author

Correction and answers

Reviewer #3

Specific comments

Line 6: delete; beef

Line 20: We have deleted "beef"

Line 10: delete; and

Line 20: We have deleted "and"

Line 28: delete; silage

Line 28: We have deleted " silage"

Line 187: change; ‘took place’ to ‘conduct’

Line 187: we have changed ‘took place’ to ‘conduct’

Line 199-202; ‘This practice is explained-’ What practice?? Please specify, it is difficult to understand to what the author refer.

We have explained "Rehydrated corn grain silage" practice …

Line 203; ‘These results are expected for rehydrated corn grain silages’ - what results? Please specify.

Line 212-213; "The improve of the silage quality using rehydrated corn grain silages is expected"

Line 205; change ‘moisture activity is necessary’ to ‘correct moisture content is’

Line 2220; We have changed ‘moisture activity is necessary’ to ‘correct moisture content is’

Line 216; change ‘silage’ to ‘ensiling’

Line 232; We have changed ‘silage’ to ‘ensiling’

Line 239; change ‘ alcohols and N-NH3’ to ‘alcohols, N- NH3’

Line 255; We have changed ‘ alcohols and N-NH3’ to ‘alcohols, N- NH3’

Line 244; ‘Overall, both silages showed’ please specify what silages

Line 244; This was corrected.

Line 269; ‘Probably the lack of effect’ please specify what effect

Sincerely yours,

Leilson Rocha Bezerra

Reviewer 4 Report

Comments and Suggestions for Authors

Title: Effects of Additives on Silage Quality of Rehydrated Corn Grains for Ruminants

provides a concise overview of the study, outlining its objectives, methodology, and key findings. Overall, the paper seems to address a relevant topic within the field of animal nutrition and silage production. It effectively introduces the research question, describes the methodology, and summarizes the main findings. The language used is clear and concise. objectives are relevant and addresses an important aspect of silage production. The review emphasize the importance of standardized control treatments and the evaluation of silage in beef animals. This provides transparency regarding the selection of studies, which is essential for the validity of the review. the main findings of the study, stating that the use of additives did not significantly influence various silage quality parameters. Overall the study is important however, the figures should be included and add more text in the paper with proper conclusion and future perspectives. 

References:

It is evident that only 32 references have been cited in this review. A paper with such a limited number of references falls short of the requirements expected of a review article. This could suggest that the authors may not have compiled a comprehensive list of pertinent studies, or it may indicate that the subject matter is not yet sufficiently developed or of significant scientific importance on a global scale. Alternatively, there is the possibility that this topic may gain importance in the future.

I would advise the authors to hold off on submitting the manuscript until they have amassed a more substantial body of references, ideally at least 100, pertaining to the topic. As a result, I recommend rejecting the manuscript in its current form.

Author Response

Response:

Dear Reviewer

We have thanked the adjustments suggested by the Reviewers of our paper, and we really appreciate the attention and your contribution in the analysis and correction of this paper. We have improved the paper from the Major Comments of the Reviewer #4 became the paper better understandable and reading. All corrections were addressed, as you can see below and in the attached file. Answers to the questions are provided below, and all the changes in the manuscript have been highlighted in red.

We have understood that the changes do not guarantee the acceptance of the manuscript although we have already been grateful for them excellent collaborations.

Thank you. We included a more significant number of articles in the Review and thus increased the discussion and information on the topic. Furthermore, the conclusion section was rewritten based on the findings of the Literature. Now, instead of 32, we have 72 references in the Review.

Sincerely yours,

Leilson Rocha Bezerra